# Photoluminescence Spectroscopy of the InAsSb-Based p-i-n Heterostructure

**DOI:** 10.3390/ma15041419

**Published:** 2022-02-14

**Authors:** Tristan Smołka, Marcin Motyka, Vyacheslav Vital’evich Romanov, Konstantin Dmitrievich Moiseev

**Affiliations:** 1Department of Experimental Physics, Faculty of Fundamental Problems of Technology, Wroclaw University of Science and Technology, Wybrzeże Wyspiańskiego 27, 50-370 Wrocław, Poland; tristan.smolka@pwr.edu.pl; 2Ioffe Institute, Politekhnicheskaya Street 26, 194021 St. Petersburg, Russia; slavutich76@mail.ru (V.V.R.); mkd-2006@mail.ru (K.D.M.)

**Keywords:** photoluminescence, spectroscopy, heterostructures, FTIR, Auger, CHHS, CHCC MOVPE, InAsSb, p-i-n, LED

## Abstract

Photoluminescence in a double heterostructure based on a ternary InAsSb solid solution was observed in the mid-infrared range of 2.5–4 μm. A range of compositions of the InAs_1−y_Sb_y_ ternary solid solution has been established, where the energy resonance between the band gap and the splitting-off band in the valence band of the semiconductor can be achieved. Due to the impact of nonradiative Auger recombination processes, different temperature dependence of photoluminescence intensity was found for the barrier layer and the narrow-gap active region, respectively. It was shown that efficient high-temperature photoluminescence can be achieved by suppressing the nonradiative Auger recombination (CHHS) process. Increased temperature, for which the energy gap is lower than the split-off band energy, leads to violation of the resonance condition in narrow gap antimonide compounds, which explains the observed phenomenon. This finding might influence future application of the investigated material systems in mid-infrared emitters used for, e.g., optical gas sensing.

## 1. Introduction

Nonradiative Auger recombination (AR) processes are multiparticle transitions of electrons and holes to excited states within the same energy band and/or to states of another energy band due to exchange of energy and momentum between carriers of charge participating in those processes [1,2]. There are two main AR mechanisms in narrow-gap semiconductors: CHCC (conduction-heavy hole-conduction-conduction) and CHHS (conduction-heavy hole-heavy hole-spin-off band) processes [3]. The CHCC process as a non-threshold AR channel was discussed elsewhere [4]. It corresponds to a small momentum transferred via the Coulomb interaction of particles, with a neglected spin-orbit interaction. In contrast, the CHHS process can be considered as a threshold-induced AR channel caused by a presence of a spin-orbit split band (SO) band. This process is possible when energy of the radiative transition equals or exceeds the split-off energy of the SO band. Therefore, to suppress the CHHS process, it is necessary to reverse the ratio of involved energies. This can be achieved by changing the band gap of a semiconductor compound by varying its composition. It is clear that a change in the composition of the solid solution will affect the entire energy band structure of the semiconductor and, therefore, the energy position of the SO band will also be shifted [5]. An example of such a compound is an InAs_1−y_Sb_y_ ternary solid solution, in which the band gap decreases from 0.4 eV to 0.1 eV [6]. A purpose of the current contribution is to reveal the possibility of controlling the energy band structure in narrow-gap compounds. Such studies should be valuable for development of mid-infrared emitters, in perspective of applications of such material systems in, e.g., optical gas sensing [7,8,9,10]. In our work, the main attention is paid to photoluminescent properties of a heterostructure based on bulk InAsSb compounds. The antimony-based materials were heavily studied in the past, but mostly in a form of low-dimensional structures [11,12,13], not in a bulk form. In this case, the novelty and importance of our work lies precisely in the disclosure of fundamental processes of radiative and non-radiative recombination in narrow-gap epitaxial layers based on antimonides. We present formulas that were used to calculate the band diagrams of the compounds forming the heterostructure, which agree with sufficient accuracy with experimental PL data. The results of our calculations can be further used for computation of energy diagrams of quantum-well heterostructures, after taking into account strain.

## 2. Materials and Methods

Samples under study were grown by metal-organic vapor phase epitaxy (MOVPE) on InAs(001) substrates in a horizontal reactor, with a resistive-heated substrate holder under hydrogen flow at atmospheric pressure. Before epitaxy, the substrate was annealed in an arsine atmosphere at a temperature of 610 °C for 30 min. Layer deposition was carried out on the substrates, which were heavily doped with Sn, until an electron concentration of n > 2 × 10^18^ cm^−3^ at the temperature of T = 510 °C was achieved. Technological details of the growth of InAsSbP layers have been reported elsewhere [14,15]. Three metalorganic compounds (trimethylindium (TMIn), trimethylstibine (TMSb) and tetrabutylarsine (tBAs)) and hydride gas phosphine (PH_3_) were used as sources of components required to grow epitaxial layers based on an In-As-Sb-P solid solution, while the ternary InAsSb solid solution was grown using TMIn, TMSb, and arsine hydride gas (AsH_3_). Diethylzinc (DeZn) was used as an acceptor dopant for formation of a p-InAsSbP barrier layer. At some point before the completion of the epitaxial growth of the InAsSbP layer, DeZn was added to the gas phase to form a heavily doped region (p ~ 2 × 10^18^ cm^−3^) in the upper part of the heterostructure. The doping profile in the other parts of the InAsSbP epitaxial layer can be determined as an exponential one-order decrease in the impurity concentration from the value declared above because of the diffusion of Zn deeper into the structure [16,17].

Sample A is a single heterostructure, consisting of a 1 μm-thick InAsSbP epilayer deposited on an InAs substrate. This sample was used to determine the energy gap of the InAsSbP quaternary solid solution isomorphic with the binary substrate (Figure 1a). Sample B can be considered to some extent as a sequential development of sample A. This is a double heterostructure containing a 1 µm-thick InAsSb epilayer sandwiched between two InAsSbP barrier layers. The mismatch value between the epilayer of the InAsSb ternary solid solution and the InAs matrix was found to be near 3.5 × 10^−3^ (Figure 1b). It should be pointed out that no extended dislocations network was visually observed. The bottom barrier layer based on the InAsSbP solid solution was lattice-matched to the InAs substrate, while the InAsSbP cap layer was in turn adapted to the lattice parameter of the InAsSb epilayer. A specific feature of sample B is that each InAsSbP barrier layer was obtained under technological conditions of epitaxy similar to the deposition of the InAsSbP layer in sample A. Despite the fact that the epitaxial growth conditions were the same, atomic compositions of the surfaces for the bottom and top barrier layers were significantly different [18]. The InAs_1−y_Sb_y_ epitaxial layer has the anionic part of the crystal lattice of the solid solution, where As atoms are replaced by Sb atoms. By adding antimony to the solid solution, the lattice misalignment of the epitaxial layer relative to the InAs substrate will be increased. As the 1-µm thick InAsSb layer is fully relaxed, consequently it imposes its own lattice parameter for the following InAsSbP cap layer. Then, during the deposition, the following epitaxial layer will tend to match the crystal lattice parameter with the lattice different from the InAs substrate and the deposited layer will be grown on the surface initially enriched with antimony. Consequently, this cap layer cannot have the same composition as the bottom barrier, even if the growth conditions are the same for the two InAsSbP layers.

In order to measure photoluminescence (PL) in a wide spectral range, an evacuated Bruker Vertex 80v Fourier transform (FT) spectrometer (Bruker, Karlsruhe, Germanys) operating in the step-scan mode was employed [19,20]. It was equipped with a liquid-nitrogen-cooled indium antimonide photodetector, a KBr beamsplitter and a 532 nm, 130 mW semiconductor laser diode that provided the pump beam. Phase-sensitive detection of optical response was performed using a lock-in amplifier. A similar FT-based approach has been demonstrated as an efficient tool for optical characterization of narrow-band gap materials in the mid-infrared spectral range [21,22,23,24,25].

## 3. Results

It is well known that multicomponent solid solutions in the In-As-Sb-P alloy system have a common cationic group (In) and are formed by mutual substitution of group V atoms in the crystal lattice of an epitaxial layer being grown. The quaternary InAs_1−x−y_Sb_y_P_x_ solid solution can therefore be considered as a combination of three binary compounds (InAs, InSb and InP) [14]. To calculate energy parameters of InAs_1−x−y_Sb_y_P_x_, we have employed the formula:(1)$$A_{{\mathrm{InAs}}_{1-x-y}{\mathrm{Sb}}_{y}P_{x}}=A_{\mathrm{InAs}}\times \left(1-x-y\right)+A_{\mathrm{InSb}}\times y+A_{\mathrm{InP}}\times x-C_{\mathrm{InAsP}}\times \left(1-x-y\right)\times x-C_{\mathrm{InAsSb}}\times \left(1-x-y\right)\times y-C_{\mathrm{InSbP}}\times x\times y$$
where A_InAs/_A_InSb/_A_InP_ are parameters associated with binary compounds that form a given multicomponent solid solution, for example, the energy of the band gap (E_G_) and the electron affinity (χ) (see Table 1); and C_InAsP/_C_InAsSb_/C_InSbP_ are bowing parameters (b_G_) that take into account the non-linear deviation that occurs during the formation of ternary compounds. For the calculations of the bandgap of the InAs_1−x−y_Sb_y_P_x_ solid solution, the following values of the bowing parameters were used: b_G_(InAsSb) = 0.61 eV, b_G_(InAsP) = 0.20 eV and b_G_(InPSb) = 1.83 eV [26].

A room temperature PL spectrum of the sample A manifested a single pronounced emission band peaked near 0.5 eV (Figure 2). We have found a good agreement between the experimental data and calculation results for the InAs_0.40_Sb_0.19_P_0.41_ composition which was taken from the microanalysis study (see the inset in Figure 2). Thus, the room-temperature PL emission band consisted of features originating from interband radiative transitions ascribed to the bandgap of the InAsSbP epilayer. The absence of any PL response from the n^+^-InAs compound can be caused by suppression of the radiative emission due to heavy doping of the substrate.

The 17 meV discrepancy between the spectral position of the main emission band and the calculated energy gap value at 77 K can be connected to radiative recombination involving Zn acceptor states situated in the forbidden gap of the p-InAsSbP epilayer (Figure 2). The low-intensity shoulder of the PL emission band (~0.5 eV) might be corresponding to radiative transitions involving deep levels induced by structural defects which are typical for InAs_1−y−x_Sb_y_P_x_ solid solutions in a composition range of phosphorus x > 0.26 [28]. It should be noted that an approximation of the calculated data by Varshni expression [29] yields an α parameter of 0.3 meV/K that is greater than that for the InAs compound due to the quaternary solid solution having a wider bandgap than the substrate.

In sample B, the low-temperature PL spectra had the low-energy (hν_1_) emission band located in the vicinity of 0.35 eV, in addition to the high-energy (hν_2_) one detected at 0.48 eV (Figure 3). The spectral position of the emission band hν_1_ equals 0.318 eV, which is associated with a radiative recombination in the narrow gap InAs_0.95_Sb_0.05_ epilayer at room temperature. A weak intensity of the hv_1_ band at low temperatures (see the inset in Figure 3) indicates further discussed Auger processes, as well as a small photo-carriers generation caused by the limited penetration depth of the laser beam. Besides, the temperature dependence of the spectral positions related to the PL emission band hν_1_ is in good agreement with the curve calculated at high temperatures T > 140 K (Figure 4a). The fit with Varshni formula yields the parameters α = 0.275 meV/K and ß = 93 K belonging to the compound enriched with InAs [29]. In the low temperature range (T < 120 K), the hν_1_ band manifested a small deviation of 18 meV from the expected value of the band gap of the ternary InAs_0.95_Sb_0.05_ solid solution. This can be connected to the transitions in the forbidden band of the narrow-gap layer involving states induced by structural defects. Initially, in the low temperature range, radiative transitions occurred between the conduction band states and acceptor states in the forbidden band; then the transformation to interband radiative transitions happened due to the thermal ionization of these impurity states with increasing temperature.

The spectral position of the high-energy emission band hν_2_ can be attributed to radiative transitions in the top barrier layer of the heterostructure under study. This PL band was observed near 0.448 eV at room temperature, which is quite close to the energy gap value for the InAs_0.40_Sb_0.22_P_0.38_ layer calculated according to Formula (1). It should be noted that the temperature dependence of the PL data in the high temperature range (T > 180 K) indicated good approximation by fit of Varshni formula with the parameters α = 0.3 meV/K and ß = 100 K, which were similar to those obtained for the InAsSbP epilayer in the sample A. The energy deviation as large as 40 meV (T = 10 K) from the band gap value can be explained by deep structural defects due to the disturbance of the arrangement of the crystal lattice caused by the enrichment of solid solution by antimony atoms [30]. The spectral position of this PL band remained unchanged at low temperatures (T < 140 K), and it is typical for acceptor-induced states in antimonide compounds.

A striking fact was that the PL bands for sample B had different dependences of their intensity with changing temperature (Figure 3). The temperature dependence of the PL intensity for the band hν_2_ manifested its exponential decrease with increasing temperature and can be presented by an Arrhenius plot, indicating a thermally activated nonradiative recombination mechanism (Figure 4b). This temperature dependence can be expressed by the following formula:(2)Iint(1T)=I0(1+B1∗e(−Ea1/kBT)+B2∗e(−Ea2/kBT)),
where B_i_ is the ratio of the radiative to nonradiative lifetimes for the *i*-th thermally activated process of nonradiative recombination at 300 K, E_Ai_ is its respective thermal activation energy, and k_B_ is the Boltzmann constant. $I_{0}$ is the initial intensity of the signal. Parameters E_A1_ = 37 meV and E_A2_ = 12 meV were calculated for the band hν_2_. It should be emphasized that the activation energy E_A1_~37 meV obtained from the Arrhenius fit is very close to the deviation in energy from the bandgap value of the top barrier layer. The low-temperature part (T = 10–120 K) of the PL intensity curve for the band hν_1_ is also described with similar behavior to the band hν_2_. Therefore, one can assume that the nature of the radiative transitions of the band hν_1_ at low temperatures and the band hν_2_ in the whole temperature range is similar. However, the high-temperature part (T = 100–300 K) of the PL intensity curve for the band hν_1_ reveals the opposite trend (sharp rise) in the temperature dependence, contrary to the hν_2_ behavior (see Figure 3 and Figure 4b). It should be noted that the intensity of the band hν_1_ exceeded the intensity of the band hν_2_ at the highest temperatures (T > 220 K). This indicates the existence of additional recombination channels involved in the PL process.

To understand the obtained results and identify possible channels of radiative and nonradiative recombination, let us turn to a schematic energy band profile of sample B, depending on the sequence of the deposited layers (Figure 4a). The InAs_0.40_Sb_0.19_P_0.41_ bottom barrier layer, which is isomorphic with the InAs substrate, forms a type II staggered heterojunction at the n-InAs/n-InAsSbP heteroboundary [31]. The narrow-gap InAs_0.95_Sb_0.05_ epilayer establishes a type-I nested heterojunction with the bottom barrier layer that provides potential confinement for charge carriers in both the conduction and valence bands of the active region, simultaneously. As shown in [32], during MOVPE growth, the matrix surface used for epitaxial deposition significantly affects the structural characteristics of the grown epilayers. Therefore, the change in the Sb fraction must result in a modification of the composition of P and/or As, while conserving the total solid phase (As + Sb + P = 1). In our case, we assume that the 1 μm InAsSb epilayer is thick enough to have its own matrix surface. As a result, a transformation of the composition of the InAsSbP quaternary solid solution from the bottom InAs_0.40_Sb_0.19_P_0.41_ barrier layer to the top InAs_0.40_Sb_0.22_P_0.38_ can be expected and observed in the PL spectra of the sample B. The InAs_0.95_Sb_0.05_/InAs_0.40_Sb_0.22_P_0.38_ heteroboundary is therefore perceived as the type-I heterojunction with a negligible energy offset in the valence band and high potential confinement for electrons in the conduction band.

Sample B represents a typical structure of a light-emitting device, with a narrow-gap active region situated between two wide-gap n- and p-InAsSbP barrier layers. No additional doping of the bottom n-InAsSbP barrier layer and the n-InAsSb active region was performed. Doping of the top barrier layer of p-InAsSbP was realized in the same manner as in the epitaxial growth of sample A. As was shown in [33], in heterostructures based on the ternary InAs_1−y_Sb_y_ solid solution with antimony content less than y < 0.1, Zn atoms can diffuse within the narrow gap layer. Thus, the active region of sample B can contain both doped (p) and compensated (i) parts in addition to the undoped n-type one. Here, we can talk about a “p-i-n”-like structure (Figure 5b).

In thermodynamic equilibrium, photoexcited electrons are moved by the built-in electric field of the p-n junction into a region of the narrow-gap layer and accumulate there due to electron confinement in the conduction band at the n-InAs_0.40_Sb_0.19_P_0.41_/n-InAsSb heterointerface. In turn, holes photogenerated in the n-InAsSb region immediately go up due to the built-in field of the p-n junction and are collected in the p-InAsSb region near the p-InAsSb/p-InAs_0.40_Sb_0.22_P_0.38_ heterointerface. As a result, the formation of a virtual dipole structure is observed, which promotes the circulation of charge carriers in the volume of the narrow-gap layer and their retention in the active region of the heterostructure. A signal from the bottom layer and substrate is not seen in the PL probably due to a confining barrier for electrons at the n-InAsSbP/n-InAsSb heterointerface formed by conduction band bending and absorption of output PL signal of the n-InAsSbP layer in the InAsSb layer.

As was mentioned above, there are two main mechanisms of nonradiative recombination in the narrow-gap semiconductors (CHCC and CHHS processes), through which charge carriers can leave the emission region. Concerning the CHCC process, because of the nonzero probability of electron transitions to high-energy states inside the conduction band, for instance by means of absorption of the PL output radiation, charge carriers can escape from the area where they were generated. This mechanism is very sensitive to a carrier concentration in the conduction band and is influenced by the strength and impact area of the built-in electric field of the heterostructure. In sample B (Figure 5b), the photoexcited electrons that originated in the p-InAsSbP compound can easily leave the top barrier layer and get to the InAsSb layer owing to a thermal ionisation mechanism of lattice phonons. Simultaneously, the Auger electrons that originated in the InAsSb layer have some difficulties in overpassing the confining barrier at the heterointerface and some concentrations of charge carriers accumulate in the narrow-gap active region. In contrast, the CHHS process mechanism predicts that holes can occupy unfilled states in the band split off the top of the valence band (Δ_SO_) on the fixed distance. This means that the energy of this Auger transition must be equal to or lower than the photon energy of the PL emission band in the narrow-gap semiconductor (Δ_SO_ < hν_PL_). The energy distance of the split-off band is determined by features of the band structure of a given solid solution. Moreover, to extract these carriers out of the split-off band the high-energy mechanism, such as avalanche or electric field ionization process, should be applied [34].

The computed estimation of the split-off band for the quaternary solid solution gives the value of Δ_SO_(InAs_0.4_Sb_0.22_P_0.38_)~0.177 eV by means of Formula (1) and use of bowing parameters b_SO_(InAsSb) = 1.2 eV, b_SO_(InSbP) = 0.75 eV and b_SO_(InAsP) = 0.16 eV [35]. The obtained value is even lower than the band gap of the quaternary solid solution at room temperature (E_G2_ = 0.448 eV). Thus, the CHHS process is still dominating in the p-InAs_0.4_Sb_0.22_P_0.38_ barrier layer and a decay of PL intensity should be observed with increasing temperature (see band hν_2_ in Figure 4b). Figure 6 shows an approximation of the experimental data of the split-off band using expression (3) as the extraction from Formula (1):(3)$$\Delta_{\mathrm{SO}}{}_{\left(\mathrm{InAsSb}\right)}=\Delta_{\mathrm{SO}\left(\mathrm{InAs}\right)}\times \left(1-y\right)+\Delta_{\mathrm{SO}}{}_{\left(\mathrm{InSb}\right)}\times y-b_{\mathrm{InAsSb}}\times \left(1-y\right)\times y,$$
where $\Delta_{\mathrm{SO}}$ is the energy difference between the valence band and the split-off band. The bowing parameter b_InAsSb_ was found to equal 1.22 eV. The obtained value well agrees with the data reported elsewhere: 1.26 eV [36] and 1.17 eV [37]. As one can see in Figure 5, there is a defined composition range of the ternary solid InAs_1−x−y_Sb_y_ solution (y < 0.2), where the band gap energy exceeds the distance to the split-off band (E_G_ ≥ Δ_SO_) at low temperatures (T = 77 K). For these compositions, the CHHS Auger process will be the dominant nonradiative recombination mechanism influencing the output radiation emission. This ratio changes in the opposite direction with increasing temperature, and we found that E_G_ < Δ_SO_ at T > 120 K. Thus, the CHHS process can be suppressed if the quantity of energy required for transition of holes to the splitting-off band is not enough (hν_1_ < Δ_SO_). With suppression of this Auger process, the probability of overlap of wave functions for electrons and holes localized in the narrow-gap layer, as well as the matrix element of recombination, will increase. Therefore, the behavior of the PL intensity curve for the hν_1_ band at high temperatures (T > 120 K) must demonstrate a sharp growth with increasing temperature. A simultaneous analysis of the hν_1_ intensity rise as well as the hν_2_ intensity decrease for temperatures higher than 120 K (where Eg < SO energy) allows us to state that related intensity change rates are significantly different. It means that hν_1_ signal increase is indeed supported by carriers activated in the barriers, but also by additional carriers, which do not contribute any more to the nonradiative Auger processes.

The split-off band energy of Δ_SO_ = 0.358 eV for the ternary InAs_0.95_Sb_0.05_ solid solution was obtained according to the calculated curve (solid square in Figure 6). Due to the split-off band indicating weak temperature dependence for narrow-gap compounds based on InAs-rich solid solutions, [39] we can assume that the bandgap of the InAs_0.95_Sb_0.05_ epilayer will be equal in energy to the split-off band for a temperature close to 120 K. That means that the resonance condition E_G_ = Δ_SO_ may occur at T = 120 K (see the crossing of the dotted line and the solid line in Figure 4a). Taking into account the temperature at which an increase in the PL intensity of the hν_1_ emission band was observed (T~140 K in Figure 4b), we assume that the energy of the resonance of the band gap and the split-off band was overcome (E_G_ < Δ_SO_). Following our calculations for the split-off band, the InAs_1−y_Sb_y_ ternary compounds will be attractive for LEDs operating in the mid-infrared range λ > 3.5 µm at high temperatures. However, the antimony composition greater than y > 0.1 for the 1 µm-thick layer of the active region suggests a rather strong mismatch of the epitaxial heterostructure with respect to the substrate matrix, which consequently increases strain and worsens structural quality. To avoid the adverse effect on the efficiency of the output radiation, the strain-compensated nanoscale structure should be considered.

## 4. Conclusions

PL spectra for the double heterostructure based on the ternary InAsSb solid solution were obtained at the temperature range of T = 10–300 K. During analysis of obtained spectrum (1) a strong rise in the PL intensity for the low-energy emission band was found with increasing temperature, (2) conditions for achieving the energy resonance in the bandgap/splitting-off band ratio were discussed, (3) it was shown that non-radiative Auger recombination process in the ternary InAs_1−y_Sb_y_ solid solutions for the composition range y < 0.1 can be suppressed at high temperatures. In summary, the current investigation manifests a way to develop narrow-gap heterostructures based on ternary InAsSb solid solutions used as an active region for light-emitting devices operating in the mid-infrared range λ > 3.5 µm around room temperature. This is important considering many applications in free space communication, environmental protection and medical diagnostics.

## Figures and Tables

**Figure 1 materials-15-01419-f001:**
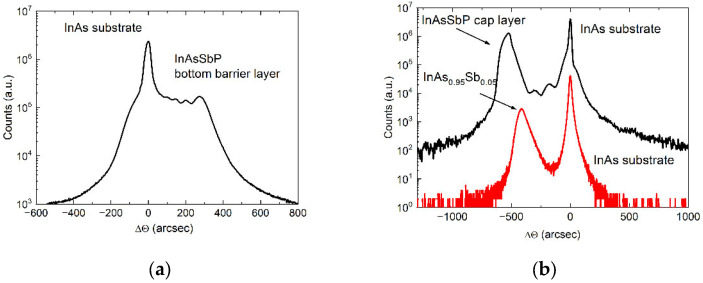
XRD pattern curves for sample A (**a**) and some parts of sample B grown on the InAs substrate (**b**).

**Figure 2 materials-15-01419-f002:**
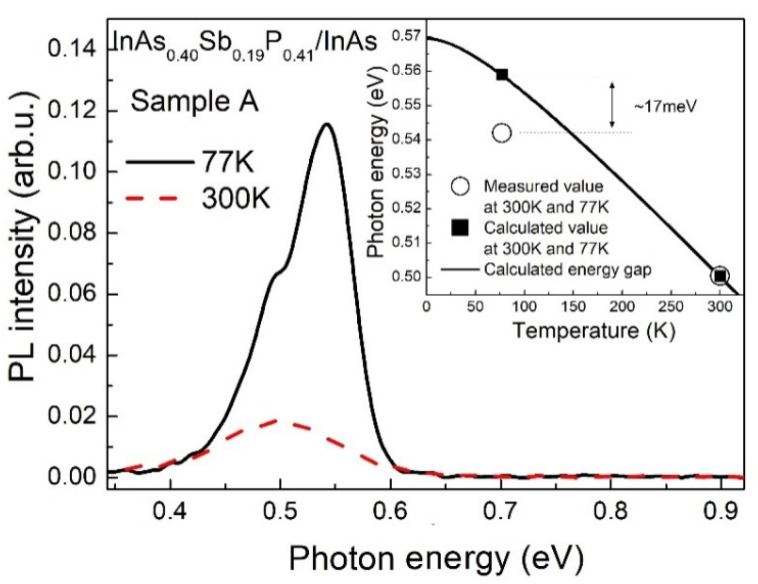
PL spectra for sample A at 77 K (solid line) and 300 K (dashed line). Inset: The temperature dependence of the energy gap of the solid solution p-InAs_0.40_Sb_0.19_P_0.41_ (experimental points—open circles, calculation data—solid squares) was obtained as a result of the Varshni approximation with parameters α = 0.3 meV/K and ß = 100 K.

**Figure 3 materials-15-01419-f003:**
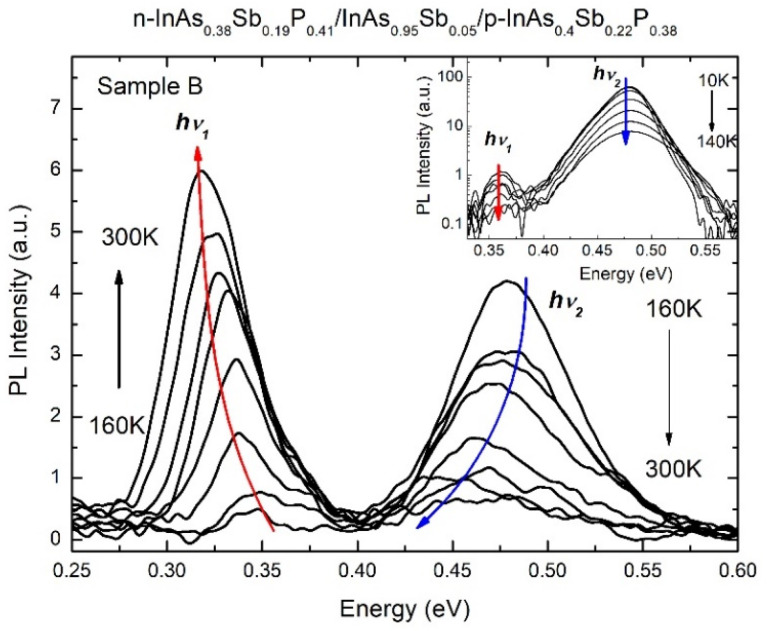
Temperature dependence (160–300 K) of PL signals for sample B. Inset: Temperature dependence of the PL signals for temperature changes in the range T = 10–140 K, where both signals decrease.

**Figure 4 materials-15-01419-f004:**
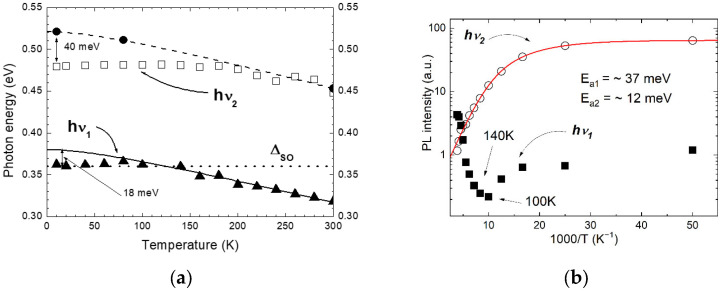
(**a**) Temperature dependences of PL band spectral positions for the InAs_0.95_Sb_0.05_ narrow-gap layer (solid triangles) and the InAs_0.40_Sb_0.22_P_0.38_ top barrier layer (open squares) of the InAsSbP/InAs_0.95_Sb_0.05_/InAs_0.40_Sb_0.22_P_0.38_ heterostructure. Solid circles are the reference points of the band gap of the InAs_0.40_Sb_0.22_P_0.38_ solid solution obtained from formula (1). Dashed curve—$E\left(T\right)=0.518\;–\;0.3*10^{-4}{*T}^{2}/\left(T+100\right)$, solid curve—$E\left(T\right)=0.38\;–\;0.275*10^{-4}{*T}^{2}/\left(T+93\right)$. (**b**) Integrated PL intensity for both (hν_1_ and hν_2_) signals measured for sample B in a function of inversed temperature. The red curve represents the best fit of the Arrhenius plot and consists of two activation energies for the band hν_2_.

**Figure 5 materials-15-01419-f005:**
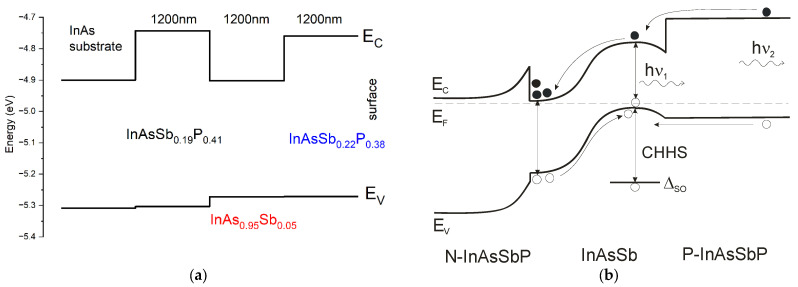
(**a**) Schematic layered energy band profile in the temperature of 77 K for sample B calculated according to Formula (1) and data from Table 1. (**b**) Energy band diagram of the n-InAsSbP/InAsSb/p-InAsSbP heterostructure at thermodynamic equilibrium. PL emission bands (hν_i_): the narrow-gap InAsSb region (1) and the p-InAsSbP barrier layer (2). The splitting-off band energy (Δ_SO_) of the narrow-gap InAs_0.95_Sb_0.05_ solid solution. CHHS is Auger non-radiative transitions in the InAsSb solid solution.

**Figure 6 materials-15-01419-f006:**
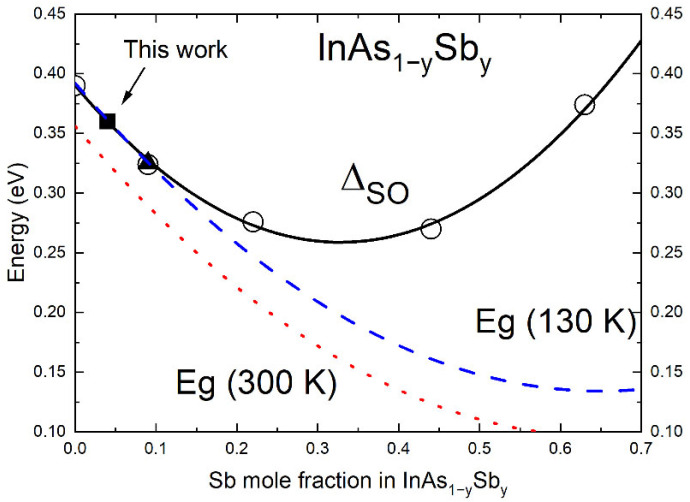
Calculated dependence of the splitting-off band (solid black line), the bandgap width at T = 130 K (dashed blue line) and T = 300 K (dotted red line) for the InAsSb ternary solid solution as a function of Sb composition. Solid triangle is from data reported in [36], open circles are experimental data from [38].

**Table 1 materials-15-01419-t001:** The band gap (E_G_) and electron affinity (χ) values of binary compounds used in the formation of a solid solution of InAs_1−x−y_Sb_y_P_x_ [27].

Binary Compound	E_G_, eV (300 K)	E_G_, eV (77 K)	E_G_, eV (4 K)	χ, eV	Δ_SO_, eV
InAs	0.354	0.408	0.417	−4.9	0.39
InSb	0.175	0.225	0.235	−4.59	0.81
InP	1.344	1.414	1.421	−4.38	0.11

## Data Availability

Not applicable.

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
