# Peer review of "Photoluminescence Spectroscopy of the InAsSb-Based p-i-n Heterostructure"

_materials, 2022, doi:10.3390/ma15041419_

Round 1
Reviewer 1 Report
Tristan SmoÅ‚ka, submitted the paper entitled “Photoluminescence spectroscopy of the InAsSb-based p-i-n heterostructure” to publish in “Materials (I.F= 3.623)”. Based on the revision performed by authors, this manuscript can be accepted after addressing the following issues.
- Improve the explanation for Equations (1), (2) and (3).
- Remove the reference [33] in conclusion section. Also provide the merits in points such as (1), (2), (3) etc., author must enhance the conclusion points.
Author Response
The answers provided inside attached document.

Reviewer 2 Report
Solid state physics is actively investigating ternary compounds and will then move to even more components. The present authors discuss photoluminescence in the MIR, a valid research topic. Besides reporting on the research, a manuscript aiming at publication ought to be comprehensible to the prospective reader.
Another reviewer has instigated improvements already, to which the authors claim that they improved the text, including their use of English. Unfortunately, the latter step has not been carried out sufficiently. Many languages differ from English in the use of articles (a/the or none), and it hinders easy reading if the wrong choice was made, as happened in many locations in the manuscript.
The sentence (in the abstract)
"This phenomenon could be explained by the fact, that when the resonance condition in narrow-gap antimonide compounds is violated by increased temperature (EG < ΔSO). " is phrased in a way which confuses me so much that I cannot even try to suggest how to make it physically sound.
I trust that in both languages the authors probably are more fluent in (Polish and Russian), the content could be expressed correctly and then translated into English without losing the meaning.
Most (not all) references are cited after the fullstop that ends a sentence. Why?
The English language flow improves after the introductory part, presumably because the authors are more familiar with laboratory talk than with free text. The experiment appears to have been executed competently. Some of the effects discussed are small, and it takes self-confidence to extract the physical information that the authors discuss. However, that is fine with me. I see no major point missing from the treatment.
Author Response

(The authors gave the same response as above.)
